# Dynamics Modeling of Industrial Robotic Manipulators: A Machine Learning Approach Based on Synthetic Data

**Sandi Baressi Šegota [1], Nikola Anđelić [1,*], Mario Šercer [2] and Hrvoje Meštrić [3]**

[1] Faculty of Engineering, University of Rijeka, Vukovarska 58, 51000 Rijeka, Croatia; sbaressisegota@riteh.hr

[2] Development and Educational Centre for the Metal Industry—Metal Centre Čakovec, Bana Josipa Jelačića 22 D, 40000 Čakovec, Croatia; mario.sercer@mev.hr

[3] Ministry of Science and Education, Donje Svetice 38, 10000 Zagreb, Croatia; hrvoje.mestric@mzo.hr

[*] Correspondence: nandelic@riteh.hr; Tel.: +385-51-505-716

**Abstract:** Obtaining a dynamic model of the robotic manipulator is a complex task. With the growing application of machine learning (ML) approaches in modern robotics, a question arises of using ML for dynamic modeling. Still, due to the large amounts of data necessary for this approach, data collection may be time and resource-intensive. For this reason, this paper aims to research the possibility of synthetic dataset creation by using pre-existing dynamic models to test the possibilities of both applications of such synthetic datasets, as well as modeling the dynamics of an industrial manipulator using ML. Authors generate the dataset consisting of 20,000 data points and train seven separate multilayer perceptron (MLP) artificial neural networks (ANN)—one for each joint of the manipulator and one for the total torque—using randomized search (RS) for hyperparameter tuning. Additional MLP is trained for the total torsion of the entire manipulator using the same approach. Each model is evaluated using the coefficient of determination ($R^2$) and mean absolute percentage error (MAPE), with 10-fold cross-validation applied. With these settings, all individual joint torque models achieved $R^2$ scores higher than 0.9, with the models for first four joints achieving scores above 0.95. Furthermore, all models for all individual joints achieve MAPE lower than 2%. The model for the total torque of all joints of the robotic manipulator achieves weaker regression scores, with the $R^2$ score of 0.89 and MAPE slightly higher than 2%. The results show that the torsion models of each individual joint, and of the entire manipulator, can be regressed using the described method, with satisfactory accuracy.

**Keywords:** industrial robot dynamics; machine learning; synthetic dataset generation

**MSC:** 68T40

## 1. Introduction

Dynamic modeling of industrial robotic manipulators is one of the key steps in industrial robotic manipulator design. In addition, it is key in various other applications such as path planning and optimization [1]. Precise dynamics models are commonly needed for any research concerning the realistic movement of industrial robotic manipulators. Still, the process of determining the dynamics model of a robot manipulator can be complex, and error-prone, exacerbated by the issue of dynamic models of individual robotic manipulators rarely being readily available to researchers. Plancher et al. (2021) [2] discussed the application of various optimizations for different hardware architectures, including CPU, GPU, and FPGA, in order to accelerate the calculation of dynamic gradients. Some authors have used artificial intelligence (AI) techniques to assist in determining the dynamic properties of a robot. For example, Yovchev and Miteva (2021) [3] presented the application of a genetic algorithm for determination of the dynamic parameter estimation, while Mitsioni et al. (2021) [4] demonstrated the application of LSTM networks to determine the dynamics of a single-action robot, namely, in the task of food cutting. There seems to be a lack of papers in recent years focusing on dynamic model generation using the machine learning approach.

Machine learning (ML) is a field within AI that allows for the creation of data-driven models. The models achieved with ML tend to be very precise, but their main pitfall is the need for large amounts of data to achieve not only precise models but models that generalize well. For this reason, a number of researchers have lately focused on synthetic dataset generation [5–7]. Synthetic dataset generation refers to the process of in silico dataset generation, where computer models are used for the generation of data points. This approach has a few benefits. Synthetic data generation can be used in instances where there is a limited number of data points that can be collected, which is extremely common when ML is applied in healthcare [8], where the patients exhibiting data belonging to a certain class may be rare [9]. Another instance where synthetic data generation may be utilized is for those cases where data collection may be extremely time-intensive [10]; this is a common application in engineering [11] and physics [12,13], as simulations in those fields may take a long time, but can be significantly sped up using simulations in high-performance computing environments. The final application of synthetic dataset generation is when measurements may be hard or expensive to perform, and the virtual generation of data points can serve to alleviate those concerns [14]. Robotics are mainly affected by the last two points, as more complex simulations may be time-intensive and require fairly expensive equipment, in the shape of the robotic manipulators themselves, as well as sensors to be locked-up in the experiment for a long time [15].

In this paper, the authors aim to apply ML with a synthetic dataset on the problem of dynamic modeling. The goal of the paper is to serve as a proof-of-concept in two areas: the first is the utilization of the synthetically generated data in machine learning within robotics; the second is the use of ML models for determination of the dynamic models of robotic manipulators. The novelty presented by this paper is also two-fold, as there is a lack of similar research in both the modeling of dynamic models using regressive ML methods, and the creation and application of synthetic datasets for the given purpose. These contributions may allow researchers to simplify the process of dynamic modeling, or modeling in general, provided they have means to collect or synthetize the data. The paper first discusses the usual process of dynamic modeling, followed by how those results have been applied to generate the dataset, with ML methodology finally being discussed-with the achieved results presented.

## 2. Materials and Methods

In this section, the methods used to generate the dataset are described along with the ML methodology used by researchers, including the algorithm itself as well as the evaluation process.

### 2.1. Dynamics Modeling

The dynamic model of the robot can be determined in various ways. In this paper, two methods were applied—the Newton–Euler (NE) algorithm and Lagrange–Euler (LE) algorithm. Two separate algorithms were used to cross-reference the obtained values and assure that the obtained model is correct. The following subsections will first present the kinematic model, which is necessary in both of the methods discussed, followed by the description of both algorithms.

The calculations and modeling were performed using the industrial robotic manipulator ABB IRB 120 manufactured by ABB Inc. (Zurich, Switzerland) for the basis of the calculations, with the measurements used (distance between joints, centers of mass, tensors of inertia) being determined using a manufacturer-provided CAD model [16]. The visualization of the used manipulator is provided in Figure 1.

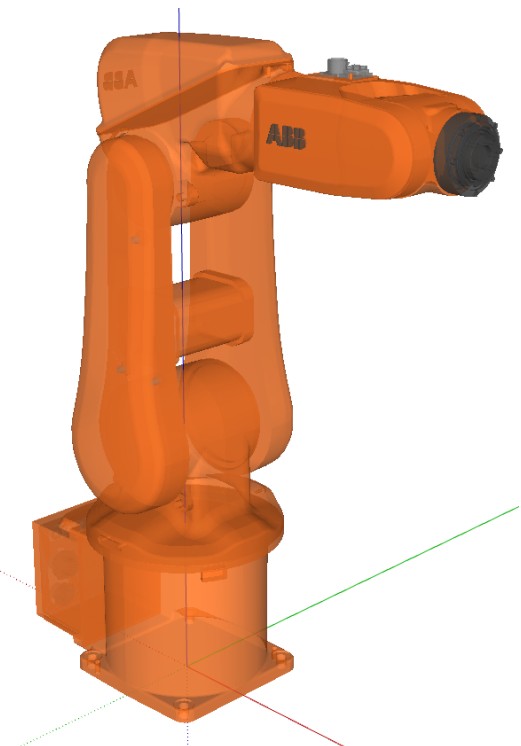

**Figure 1.** The modeled robotic manipulator ABB IRB 120 in isometric view [16].

### 2.1.1. Kinematic Model

The Dennavit–Hartenberg (DH) method was applied by setting an orthonormal coordinate system in each of the robotic manipulator joints, where axis $z$ matches the axis of the joint. With the coordinate systems positioned, the parameters $\theta_k$, $\alpha_k$, $d_k$, and $a_k$ can be determined based on the distances between the centers of the coordinate systems and the relations between the axis [17]. The values can then be placed into a transformation matrix. The transformation matrix between joints $k-1$ and $k$ is given as [18]

$$T_{k-1}^k = \begin{bmatrix} cos(\theta_k) & -cos(\alpha_k) \cdot sin(\theta_k) & sin(\alpha_k) \cdot sin(\theta_k) & \alpha_k \cdot cos(\theta_k) \\ sin(\theta_k) & cos(\alpha_k) \cdot cos(\theta_k) & -sin(\alpha_k) \cdot cos(\theta_k) & \alpha_k \cdot sin(\theta_k) \\ 0 & sin(\alpha_k) & cos(\alpha_k) & d_k \\ 0 & 0 & 0 & q \end{bmatrix} \tag{1}$$

The transformation matrix of the entire manipulator is calculated using a product of all the individual joint transformation matrices [19]:

$$T_{base}^{tool} = \Pi_{k-1}^k T_{k-1}^k, \tag{2}$$

resulting in a matrix given as [20]:

$$T_{base}^{tool}(q) = \begin{bmatrix} R(q) & p(q) \\ v_1^T & \sigma \end{bmatrix}, \tag{3}$$

where the tool orientation matrix $R(w) = [r^1 r^2 r^3]$ consists of the perpendicular vector $r^1$, movement vector $r^2$, and approach vector $r^3$ and $p(q)$ represents the tool end position. Values $v_1^T$ and $\sigma$ represent the perspective vector and the scaling coefficient, commonly set to $[000]$ and 1, respectively [21]. The calculated transformation matrices can then be used within the NE and LE algorithms.

2.1.2. Lagrange–Euler

The basis of the LE algorithm is the definition of the differential equations that serve to calculate the torque of joint $i$ as $\tau_i$ using [22]

$$\tau_i = \sum_{j=1}^{n}[D_{ij}(q)q_j] + \sum_{k=1}^{n}\sum_{j=1}^{n}[C_{kj}^i(q)q_kq_j] + h_i(q) + b_i(q), \tag{4}$$

where $\sum_{j=1}^{n}[D_{ij}(q)q_j]$ defines the moments and the inertial forces, Coriolis forces are presented by the term $\sum_{k=1}^{n}\sum_{j=1}^{n}[C_{kj}^i(q)q_kq_j]$, gravity's effect is given by $h_i(q)$, and $b_i(q)$ defines the internal friction of the manipulator's joint.

In the beginning of the LE algorithm, three values are defined. First is the iterator $i$ set to 1, followed with $T_0^0$, a $4 \times 4$ identity matrix, and $D(q)$, a $3 \times 3$ zeroes matrix. The LE algorithm then starts by calculating the tensor of inertia $D_{i'}(q)$ with [23]

$$\begin{aligned} D_{i'}(q) &= \begin{bmatrix} I_{xx} & I_{xy} & I_{xz} \\ I_{yx} & I_{yy} & I_{yz} \\ I_{zx} & I_{zy} & I_{zz} \end{bmatrix} \\ &= \begin{bmatrix} \int_{V_k}(y^2 + z^2)\rho dV & -\int_{V_k}xy\rho dV & -\int_{V_k}xz\rho dV \\ -\int_{V_k}xy\rho dV & \int_{V_k}(y^2 + z^2)\rho dV & -\int_{V_k}yz\rho dV \\ -\int_{V_k}xz\rho dV & -\int_{V_k}yz\rho dV & \int_{V_k}(y^2 + z^2)\rho dV \end{bmatrix} \end{aligned} \tag{5}$$

Following this, the vector $z$ for the joint $i-1$ is calculated as per [24]

$$z^{i-1}(q) = R_0^{i-1}(q) \cdot i^3, \tag{6}$$

followed by the calculation of the homogeneous transformation between the base and the current joint [25]:

$$T_0^i = T_0^{i-1}(q)T_{i-1}^i(q). \tag{7}$$

To transpose the position of the center of mass in relation to the coordinate system of the base, the following equation is used [26]:

$$c^i(q) = \begin{bmatrix} 1 & 0 & 0 & 0 \\ 0 & 1 & 0 & 0 \\ 0 & 0 & 1 & 0 \end{bmatrix} T_0^i(q)\delta c^i, \tag{8}$$

with $\delta c^i$ being the homogeneous coordinates of the robotic link $i$. The tensor of inertia in relation to the base coordinate system can then be calculated:

$$D_i(1) = R_0^i(q)D_{i'}(q)[R_0^i(q)]^T. \tag{9}$$

To correlate the infinitesimal movements of the manipulator joints and the infinitesimal movements of the tool, the Jacobian matrix is defined [27]:

$$J^i(q) = \begin{bmatrix} A^k(q) \\ B^k(q) \end{bmatrix} = \begin{bmatrix} \frac{\partial c^i(q)}{\partial q_i} & \cdots & \frac{\partial c^i(q)}{\partial q_i} & 0 \cdots 0 \\ \theta_1 z^0(q) & \cdots & \theta_1 z^{i-1}(q) & 0 \cdots 0 \end{bmatrix} \tag{10}$$

The total torsion of inertia can be calculated with [28]:

$$D(q) = \sum_{i=1}^{n}[A^i(q)]^T m_i[A^i(q)] + [B^i(q)]^T D_i(q)[B^i(q)], \tag{11}$$

with $m_i$ being the mass of the current joint. If the tensors of inertia have not been calculated for all the individual joints, the procedure is repeated for the next joint. If the calculation

has been performed, $i$ is reset to 1, and the calculations are performed for each of the joints for the speed connectivity matrix [29]:

$$C_{kj}^i = \frac{\partial D_{ij}(q)}{\partial q_k} - \frac{\partial D_{kj}(q)}{\partial q_i}, 1 \leq i, j, k \leq n, \tag{12}$$

gravity influence vector, as per

$$h_i(q) = \sum_{k=1}^{3} \sum_{j=1}^{n} [g^k m_j A_{ki}^j(q)], 1 \leq i \leq n, \tag{13}$$

and finally, the friction is approximated using Tustin's friction model [30]:

$$b_k(\dot{q}_k) = b_k^v \dot{q}_k + sgn(\dot{q}_k)[b_k^d + (b_k^s - b_k^d)exp(\frac{-|\dot{q}_k|}{\epsilon})]. \tag{14}$$

Once the second iteration of calculations is complete, each of the joints has an equation calculated, relating to the joints torque defined using [31]:

$$\tau_i = \sum_{j=i}^{n} [D_{ij}(q)\ddot{q}_j]' \sum_{k=1}^{3} \sum_{j=1}^{n} [C_{kj}^i(q)\dot{q}_k\dot{q}_j] + h_i(q) + b_i(\dot{q}). \tag{15}$$

### 2.1.3. Newton–Euler

NE differentiates from LE in the fact that it has a forward (in the direction from the base of the manipulator to the tool) and backward (from the tool to the base of the robotic manipulator) calculation. In the forward calculation, the speeds and accelerations (linear and angular) are calculated for each joint. In the backward calculation, the forces and momenta on each of the links are calculated. At the start of the NE algorithm, initial values need to be set [22]:

- $T_0^0 = I$,
- $f^{n+1} = -f^{tool}$,
- $n^{n+1} = -n^{tool}$,
- $v^0 = 0$,
- $\frac{dv^0}{dt} = -g$,
- $\omega^0 = 0$,
- $\frac{d\omega^0}{dt} = -0$, and
- $i = 0$.

The initial calculation step is the same as in LE—determining the vector $z$,

$$z^{i-1}(q) = R_0^{i-1}(q) \cdot i^3, \tag{16}$$

followed by the calculation of the angular speed $\omega$ [32]:

$$\omega^k = \omega^{i-1} + \zeta_i \cdot \frac{dq_i}{dt} z^{i-1}(q), \tag{17}$$

with $\zeta_i$ being set to 1 for the revolutional joint and to 0 for the linear joint. The angular speed is calculated with [33]

$$\dot{\omega}^i = \dot{\omega}^{i-1} + \zeta_i [\frac{d^2 q_i}{dt^2} z^{i-1}(q) + \omega^{i-1} \times \frac{dq_i}{dt} z^{i-1}(q)]. \tag{18}$$

The complex homogeneous transformation matrix is again determined as

$$T_0^i = T_0^{i-1} T_{i-1}^k, \tag{19}$$

which allows for calculation of the vector [34]

$$\delta s^i = \begin{bmatrix} 1 & 0 & 0 & 0 \\ 0 & 1 & 0 & 0 \\ 0 & 0 & 1 & 0 \end{bmatrix} (T_0^i - T_0^{i+1}) i^4. \tag{20}$$

The final value that needs to be calculated is the linear acceleration [35]:

$$\frac{dv^i}{dt} = \frac{dv^{i-1}}{dt} + \frac{d\omega^i}{dt} \times \delta s^i + \omega^i \times (\omega \times \delta s^i) + (1 - \zeta_i)[\frac{d^2 q_i}{dt^2} z^{i-1} + 2\omega^i \times \dot{q}_i z^{i-1}]. \tag{21}$$

This process is repeated for each of the joints, until the final joint of the robotic manipulator is reached. At that point, the backward calculation begins, from the final joint to the base. The first value to be calculated is the vector $r^i$ [36]:

$$\delta r^i = \begin{bmatrix} 1 & 0 & 0 & 0 \\ 0 & 1 & 0 & 0 \\ 0 & 0 & 1 & 0 \end{bmatrix} T_0^i (\delta c^i - i^4). \tag{22}$$

The force acting on the joint $i$ is calculated using [37]:

$$f^i = f^{i+1} + m_i[\frac{dv^i}{dt} + \frac{d\omega^i}{dt} \times \delta r^i + \omega^i \times (\omega^i \times \delta r^i)]. \tag{23}$$

The momentum of the joint can consequently be calculated according to [38]

$$n^i = n^{i+1} + (\delta s^i + \delta r^i) \times f^i - \delta r^i \times f^i + R_0^i D_{i'} (R_0^i)^T \frac{d\omega^i}{dt} + \omega^i \times (R_0^i D_{i'} (R_0^i)^T \omega^i) \tag{24}$$

with $D_{i'}$ defined as per Equation (5). With the force and the momentum calculated, we can determine the joint actuator momentum using the following equation [39]:

$$\tau_i = \zeta_i (n^i)^T z^{i-1} + (1 - \zeta_i)(f_i)^T z^{i-1} + b_i(\dot{q}_i). \tag{25}$$

The value of the iterator $i$ is then lowered, and the calculation is repeated for the next joint. Once the base of the robot manipulator is reached, the NE algorithm is completed.

### 2.2. Dataset Generation

The dataset was generated by taking the equations obtained using the methods described in the previous section. As can be seen by observing Equations (15) and (25), the inputs necessary to calculate the joint torsion are the joint position $q_i$, the angular speed of the joint $\dot{q}_i$, and the angular acceleration of the joint $\ddot{q}_i$. Only the angular speeds and accelerations are considered since all the joints in the modeled robotic manipulator are rotational.

To generate the dataset, the values $[q_i \dot{q}_i \ddot{q}_i] \forall i \in [1, 6]$ are uniformly randomly generated. The value of the $\tau_i \forall i \in [1, 6]$ are then calculated using the equations obtained from the NE algorithm and verified using the LE model. The ranges of variables for random generation are set as given in Table 1. The values for the individual joints have been selected according to the ranges provided by the manufacturer [40]. Values for the minimal and maximal joint speeds and accelerations have been set uniformly for all joints, with the values selected as being realistic speeds and accelerations that could be encountered during the operation of the industrial robotic manipulator, to the ranges of $[-1, 1]$ rad/s and $[-1, 1]$ rad/s$^2$ [40].

The total torque of all the joints is calculated as the sum of all the joint torques $\tau = \sum_i = 1^n |tau_i|$ [41]. $Q$ defines all the joint position values, $\dot{Q}$ defines all the angular speeds of joints, $\ddot{Q}$ defines the angular accelerations of the joints, and $T$ are the values of

the joint torques; then, the values are written in a Comma-Separated Values (CSV) file in the following shape:

$$\begin{bmatrix} Q & \dot{Q} & \ddot{Q} & T & \tau \end{bmatrix},$$ (26)

where the input vector consists of

$$\begin{bmatrix} Q & \dot{Q} & \ddot{Q} \end{bmatrix}.$$ (27)

A total of 20,000 data points were generated in this manner. While the inputs are generated uniformly and randomly, meaning their distribution is known, the outputs may have a different distribution. For this reason, the histograms of the outputs are plotted and shown in Figure 2. The analysis of the histograms was performed through distribution fitting [42,43]; this analysis shows that the datasets generated for individual joints follow a generalized normal distribution [44] centered around 0, while the data generated for the total joint torque follows a reciprocal inverse Gaussian distribution [45].

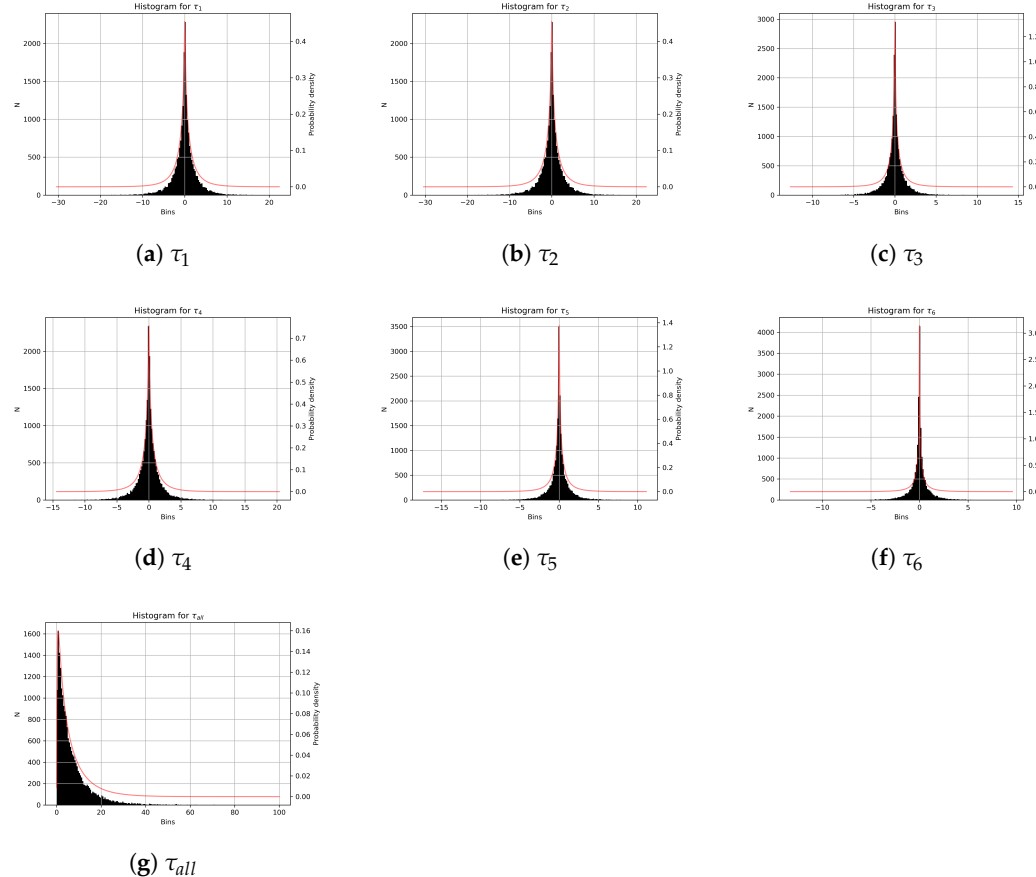

**Figure 2.** Distributions of the synthetically generated outputs: (**a**–**f**) the distributions of the generated values for individual joints; (**g**) the total torque of the robotic manipulator.

**Table 1.** The upper and lower bounds for all the randomly determined values during the process of dataset generation.

| Value | Symbol | Lower Boundary | Upper Boundary | Unit |
|---|---|---|---|---|
| Angle of joint 1 | $q_1$ | −2.88 | 2.88 | rad |
| Angle of joint 2 | $q_2$ | −1.92 | 1.92 | rad |
| Angle of joint 3 | $q_3$ | −1.22 | 1.92 | rad |
| Angle of joint 4 | $q_4$ | −2.79 | 2.79 | rad |
| Angle of joint 5 | $q_5$ | −2.09 | 2.09 | rad |
| Angle of joint 6 | $q_6$ | 0 | 6.28 | rad |
| Speeds * | $\omega_i, \dot{q}_i$ | −1.00 | 1.00 | rad/s |
| Accelerations * | $\dot{\omega}_i, \ddot{q}_i$ | −1.00 | 1.00 | rad/s$^2$ |

* For all joints, 1 through 6.

In a realistic application, the dataset would have instead been collected using a sensor array that measures the aforementioned values on a robotic manipulator. Still, in this instance, a synthetic approach was selected to test the validity of synthetic dataset generation.

*2.3. Machine Learning Approach*

The ML algorithm selected for use in the presented research is the multilayer perceptron (MLP). MLP is a feed-forward type of artificial neural network, which is trained using the processes of forward propagation and backpropagation.

Forward propagation refers to the process used by the MLP model to obtain the output values. The model consists of neurons placed in layers, using a fully connected architecture in which every neuron in one layer is connected to all the neurons in the subsequent layer, using weighted connections. The value of each individual neuron—barring the "input" neurons in the first layer, which are set to the values of the inputs being modeled—is calculated as the activated weighted sum of the values of neurons in the previous layer as per [46,47]

$$x_k^j = F(\sum_{i_0}^{n} w_i^{(i,j-1)\to(k,j)} \cdot x_i^{j-1}), \tag{28}$$

where $x_k^j$ is the value of a given neuron, $x_i^{j-1}$ is the value of the neuron in the previous layer, $w_i^{(i,j-1)\to(i,j)}$ is the weight of the connection between the $i$-th neuron in the layer $j-1$ and the $k$-th neuron in the layer $j$, with $F$ being the activation function—a predefined function that serves to transform the output of the neuron by either eliminating the unwanted values (ReLU) or limiting the output (sigmoid) [48].

To obtain a well-performing model, the weights connecting the neurons need to be adjusted. This is performed in the backpropagation process. When the input neurons are set to the value of desired inputs $X_i$, forward propagation is performed using Equation (28) to generate the values for each of the layers. This process is repeated until the last layer, consisting of a single neuron, is reached. The value of that neuron $\hat{y}_i$ is used as the output of the MLP model. Comparing that value with the expected output $y_i$ will yield a difference that is the error of the model for the given weight $W$, commonly referred to as the cost function, defined as [49]

$$J(W) = \frac{1}{n} \sum_{i=0}^{n} |y_i - \hat{y}_i|. \tag{29}$$

This error is then used to adjust the weights of the model using gradient-based adjustment. If we define $\alpha$ as the learning rate—the value that specifies how fast the model learns—then the weight adjustment between the new weight values in layer $j$-$W_j'$ and old values $W_j$ can be defined using [50,51]

$$W_j' = W_j - \alpha \cdot \frac{\partial J(W)}{\partial W_j}.$$ (30)

The introduced $\alpha$ is one of the so-called hyperparameters of the model. These are values that define the model architecture, and obtaining correct values of those hyperparameters is the key to obtaining a quality model. A number of hyperparameters can be tuned, and the ones that were adjusted in the presented research are as follows [52]:

- Hidden layer size—the number of neurons and layers, given as a tuple in which each value presents a number of neurons in a given layer;
- Activation function—activation function to be used within all of the model's neurons;
- Initial learning rate—the learning rate of the model;
- Learning rate type—the manner in which the learning rate is adjusted through the training process, inversely to the elapsed training iterations, kept constant, or adapted to the model error;
- Solver—the algorithm used for weight adjustment during training. The possible solvers are Adam, Stochastic Gradient Descent (SGD), and Limited-Memory Broyden–Fletcher–Goldfarb–Shanno (LBFGS);
- L2 regularization parameter—the value that controls the influence of the individual inputs, preventing a single input from having too much influence on the output, to provide models that have better generalization.

As previously mentioned, the hyperparameter tuning process is key in achieving a well-performing model. The issue is that there are no set rules as to which hyperparameters will perform well for a given problem [53]. For this reason, a randomized search (RS) is defined [52]. Possible values of the hyperparameters are either set as a list, if discrete, or given as a range if continuous, as shown in Table 2. The random search procedure then randomly constructs a vector of hyperparameter values and uses that value to construct a model that is then trained using the forward- and backpropagation process previously described. The trained model is evaluated, and the process is repeated until satisfactory scores are achieved, or the process is manually interrupted—in which case the best-achieved model is presented. The evaluation procedure used is given in the subsection below.

**Table 2.** Possible hyperparameter values. In cases where the hyperparameter is selected from the list (Hidden Layer Sizes, Activation Function, Learning Rate Type, and Solver), the possible values are given, while in the cases where the values are selected randomly from a range (Initial Learning Rate, L2 Regularization), the lower and upper bound are given.

| Hyperparameter | Values | |
|---|---|---|
| Hidden Layer Sizes | (288), (288, 288), (288, 288, 288), (288, 288, 288, 288), (288, 288, 288, 288, 288), (144), (144, 144), (144, 144, 144), (144, 144, 144, 144), (144, 144, 144, 144, 144), (72), (72, 72), (72, 72, 72), (72, 72, 72, 72), (72, 72, 72, 72, 72), (36), (36, 36), (36, 36, 36), (36, 36, 36, 36), (36, 36, 36, 36, 36), (18), (18, 18), (18, 18, 18), (18, 18, 18, 18), (18, 18, 18, 18, 18) | |
| Activation Function | ReLU, Logistic, Identity | |
| Initial Learning Rate | 0.0001 | 0.5 |
| Learning Rate Type | Constant, Adaptive, Inverse Scaling | |
| Solver | Adam, LBFGS, SGD | |
| L2 Regularization Parameter | 0.0001 | 0.5 |

It should be noted that the tuned hyperparameters, for the number of neurons per layer, only affect the so-called hidden layers between the input and output neuron layers, which are defined by the problem being modeled. While the output layer always consists of one neuron, the number of input neurons depends on the problem that is being regressed. In the presented case, the inputs consist of 18 values, these being the position of each joint, angular speed of each joint, and angular acceleration of each joint. As there are six joints present in the robot manipulator that is being modeled, with three values per joint, this means that each model will have eighteen inputs. Since each of the models can only have one input, only one value can be regressed at one time. For this reason, seven different models are developed—one for the torque of each joint and one for the total torque.

Model Evaluation

The trained models were evaluated using two metrics: coefficient of determination ($R^2$) and mean absolute percentage error (MAPE). $R^2$ compares two sets of data, the predicted values $\hat{y}$ and $y$, in terms of variance. $R^2$ is calculated using [54]

$$R^2 = 1 - \frac{\sum_{i=0}^{n}(y_i - \hat{y}_i)^2}{\sum_{i=0}^{n}(y_i - \frac{1}{n}\sum_{i=0}^{n} y_i)^2} \tag{31}$$

and its value will be 1.0 in the case when there is no unexplained variance between two sets (the desired outcome) and 0.0 when there is no explained variance between the datasets [55]. While being an effective and popular measure, $R^2$ can be hard to directly interpret. For this reason, MAPE is introduced as a secondary performance measure. *MAPE* is expressed as the percentage of the value range that the average achieved absolute error is and can be calculated using [56,57]

$$MAPE = \frac{100\%}{n} \sum_{i=0}^{n} |\frac{y_i - \hat{y}_i}{y}|. \tag{32}$$

Splitting the dataset into training and testing set in order to determine the performance is an industry-standard practice in ML. In this approach, the dataset is split into two parts, where the first part (training) is utilized in the training process described in the previous subsection, while the evaluation is performed on the testing dataset, which is data previously unseen by the model. This approach has certain issues. The main issue is that the random training–testing split can be particularly positive for the model being evaluated. This can lead to deceptively high-performance metrics for a model that happened to obtained the right data but would perform poorly in a generalized environment with new data provided to it [58]. For this reason, cross-validation was performed. Instead of splitting the dataset into training and testing sets, the dataset was split into 10 equal parts, so-called folds [59]. Then, the training–testing procedure was repeated ten times, each time with a different data fold being used as the testing dataset, with the remaining folds being used for training. The scores are then expressed as the average score across all folds, with a standard error. This allows determining the performance of the model on the entirety of the collected dataset [60].

## 3. Results and Discussion

The best-achieved results per each of the targets are given below in Table 3. The models trained using RS were set to test new hyperparameters until the $R^2$ value of 0.99 was reached, or for 10,000 iterations. None of the models achieved the $R^2$ score necessary to preemptively stop the execution and were trained for the full number of RS iterations. Observing Table 3, it can be seen that all the individual joint torque models achieved $R^2$ scores higher than 0.90 and MAPE below 2%. These scores indicate a successful regression, especially considering the relatively high number of data points and the relatively complex problem being modeled. Observing the individual joints, it can be seen that the first four joints (in the direction from the base to the tool of the robotic manipulator) achieved $R^2$

scores higher than 0.95, indicating high-quality models. All of the models exhibit very low standard deviations, indicating that they are stable across various data folds.

For the first joint model $\tau_1$, the average $R^2$ achieved across the folds is 0.96, with a standard deviation of 0.01. The model in question also achieved the lowest MAPE, with 1.18% average error across folds and a standard error of 0.03%. A relatively large neural network was used, with three hidden layers of 288 neurons activated using the logistic activation function. The learning rate was set on the lower side of the range but was adapted during the execution. The L2 regularization parameter was set high in comparison with the other models and the selected solver algorithm was Adam. Similar values were used for $\tau_2$, $\tau_3$, and $\tau_4$. Exceptions are that $\tau_2$ utilized a significantly smaller network architecture consisting of three layers of 144 neurons, achieving an $R^2$ score of 0.98 with a standard error of 0.04 and MAPE of 1.16% with a standard error of 0.02, which are the best scores achieved by any of the models on any of the joints. Observing $\tau_3$, it differs by using a neural network with an additional layer of 288 neurons, a ReLU activation function, and a constant learning rate. $\tau_3$ managed to achieve somewhat poorer, but still very good scores of $0.95 \pm 0.04$ for $R^2$ and $1.59 \pm 0.03\%$ for MAPE. Finally, $\tau_4$ achieved an $R^2$ of $0.96 \pm 0.03$ and MAPE of $1.81 \pm 0.08$, differing from its predecessors by using the inverse scaling adjustment for the learning rate.

Models for $\tau_5$ and $\tau_6$ show somewhat weaker results, with $R^2$ scores of $0.92 \pm 0.05$ and $0.93 \pm 0.03$, and MAPE scores of $1.91 \pm 0.02$ and $1.93 \pm 0.03$, respectively. The $\tau_5$ model uses an ANN architecture with four hidden layers of 144 neurons and a hyperbolic tangent activation function. The learning rate of the model is near the upper side of the range at 0.4375 and is not adjusted during the execution. The regularization parameter value was set at 0.00184—significantly lower than other models' regularization values. $\tau_6$ utilizes the smallest of all the neural networks, with two layers of 144 neurons. The same activation function was used as in $\tau_5$. This model uses a relatively high learning value but allows for its adaptation. Both $\tau_5$ and $\tau_6$ models used the LBFGS solver algorithm.

Finally, we can observe the model for the total joint torque $\tau_{all}$. This model is similar to the first four joints, with three hidden layers of 288 neurons, activated with logistic function. The inverse scaling learning rate is applied to the initial learning rate of 0.00951. Adam regularization function is used, as in the best-performing models, for the first four joints. A relatively high regularization value is used for the $\tau_{all}$ model.

For the ease of result comparison, the achieved scores per each goal are also given in Figures 3 and 4. Figure 3 shows the comparison between the achieved $R^2$ scores. The drop in performance between the first four joints, the fifth and sixth joint, and the total torque has already been noted. This is also noticeable in Figure 4, where the same trend can be noticed with the increase in the error value.

The values that determine a high-quality solution vary depending on the problem at hand. For example, models trained on larger datasets have a tendency to exhibit lower scores due to a larger amount of variance in the dataset [58]. In the presented research, due to the high number of data points and a complex problem that is attempting to be regressed (robot dynamics are described by very large mathematical models), we can consider the values of $R^2 \geq 0.9$ and $MAPE \leq 2\%$ as indicative of a high-quality model.

Observing all the values, it can be seen that the RS process led to the selection of larger network architectures. This indicates that the modeled problem is relatively complex regarding its ease-of-modeling using the MLP algorithm. Still, all the models achieved results that can be regarded as satisfactory. It is interesting to note that the poorest performing model is the only one that has a non-normal distribution, supporting a potential link to modeling complexity.

**Table 3.** The best results achieved for all the torque targets, with the model hyperparameters used in the best-performing models.

| Target | $R^2$ | $\sigma_{R^2}$ | MAPE | $\sigma_{MAPE}$ | Hyperparameters | |
|--------|-------|----------------|------|-----------------|-----------------|---|
| $\tau_1$ | 0.95774 | 0.01285 | 1.17815 | 0.03527 | Hidden Layer Sizes | 288, 288, 288 |
| | | | | | Activation | Logistic |
| | | | | | Initial Learning Rate | Adaptive |
| | | | | | Learning Rate Type | 0.00923 |
| | | | | | Solver | Adam |
| | | | | | Regularization | 0.12142 |
| $\tau_2$ | 0.98306 | 0.04280 | 1.15615 | 0.02649 | *Hidden Layer Sizes* | 144, 144, 144 |
| | | | | | Activation | Logistic |
| | | | | | Initial Learning Rate | 0.01656 |
| | | | | | Learning Rate Type | Adaptive |
| | | | | | Solver | Adam |
| | | | | | Regularization | 0.01189 |
| $\tau_3$ | 0.95162 | 0.03831 | 1.59342 | 0.03402 | Hidden Layer Sizes | 288, 288, 288, 288 |
| | | | | | Activation | ReLU |
| | | | | | Initial Learning Rate | 0.01432 |
| | | | | | Learning Rate Type | Constant |
| | | | | | Solver | Adam |
| | | | | | Regularization | 0.09456 |
| $\tau_4$ | 0.96318 | 0.03493 | 1.80749 | 0.07908 | Hidden Layer Sizes | 288, 288, 288 |
| | | | | | Activation | Logistic |
| | | | | | Initial Learning Rate | 0.00997 |
| | | | | | Learning Rate Type | Inverse Scaling |
| | | | | | Solver | Adam |
| | | | | | Regularization | 0.010375 |
| $\tau_5$ | 0.91787 | 0.04833 | 1.90698 | 0.01564 | Hidden Layer Sizes | 144, 144, 144, 144 |
| | | | | | Activation | Tanh |
| | | | | | Initial Learning Rate | 0.04375 |
| | | | | | Learning Rate Type | Constant |
| | | | | | Solver | LBFGS |
| | | | | | Regularization | 0.00184 |
| $\tau_6$ | 0.92712 | 0.02718 | 1.93007 | 0.02965 | Hidden Layer Sizes | 144, 144 |
| | | | | | Activation | Tanh |
| | | | | | Initial Learning Rate | 0.01992 |
| | | | | | Learning Rate Type | Adaptive |
| | | | | | Solver | LBFGS |
| | | | | | Regularization | 0.12729 |
| $\tau_{all}$ | 0.89479 | 0.03945 | 2.04094 | 0.02421 | Hidden Layer Sizes | 288, 288, 288 |
| | | | | | Activation | Logistic |
| | | | | | Initial Learning Rate | 0.00951 |
| | | | | | Learning Rate Type | Inverse Scaling |
| | | | | | Solver | Adam |
| | | | | | Regularization | 0.10276 |

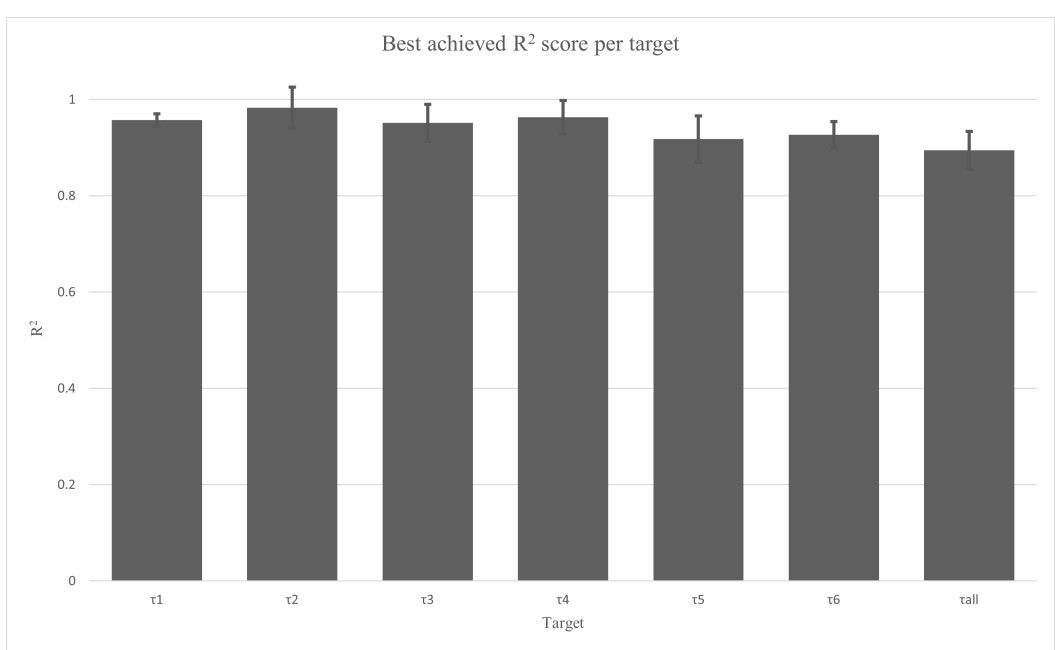

**Figure 3.** Best results achieved per goal, evaluated with $R^2$ (higher is better).

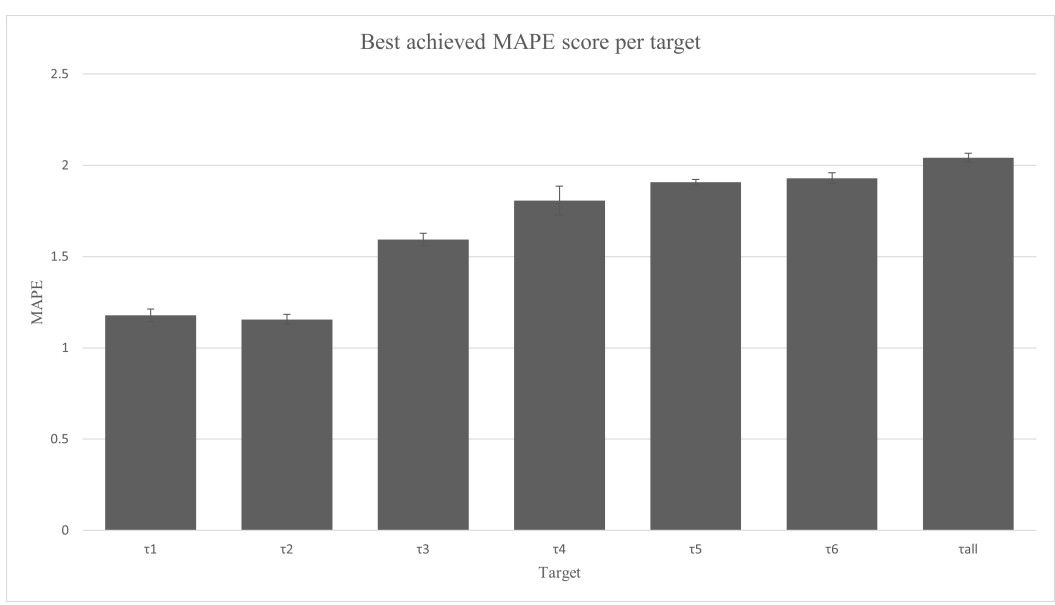

**Figure 4.** Best results achieved per goal, evaluated with MAPE (lower is better).

## 4. Conclusions

The paper presents the utilization of NE and LE algorithms for the modeling of the industrial robot manipulator dynamics. The obtained mathematical models are then used to generate a synthetic dataset used for the training of ML-based models using the MLP algorithm. The achieved results are promising and point towards two possibilities. The first is the use of ML algorithms, namely, ANNs, for the dynamic modeling of industrial robotic manipulators. It should be noted that in a realistic application scenario, the data used would be collected from sensors. This leads to the second possibility investigated in the paper—the use of the synthetically designed dataset in the area of robotics modeling, which can assist in saving time and funds during research operations.

Future work in the field could include the application of different ML algorithms with the goal of model quality improvement, and further testing on synthetic datasets in

robotics,such as investigating whether an improvement can be seen when real-world data are mixed with synthetically generated data.

The paper presents the utilization of NE and LE algorithms for the dynamics modeling process of an industrial robotic manipulator. The paper also showcases the use of the generated models in the creation of a synthetic dataset, which is used to train an ML-based MLP algorithm. The torsion values were regressed for each of the six joints, as well as the total torque. For the first joint, the MLP managed to achieve a model with scores of $R^2 = 0.96 \pm 0.01$, $MAPE = 1.18\% \pm 0.04\%$. Scores for the second joint were $R^2 = 0.09 \pm 0.04$, $MAPE = 1.15\% \pm 0.03\%$, and for the third, $R^2 = 0.95 \pm 0.04$, $MAPE = 1.59\% \pm 0.03\%$. The scores for the fourth and fifth joint were $R^2 = 0.96 \pm 0.03$, $MAPE = 1.81\% \pm 0.08\%$ and $R^2 = 0.92 \pm 0.05$, $MAPE = 1.91\% \pm 0.02\%$, respectively. The best-achieved scores for the sixth joint were $R^2 = 0.92 \pm 0.03$, $MAPE = 1.93\% \pm 0.03$. Finally, the scores for the total torque of the industrial robotic manipulator were $R^2 = 0.89 \pm 0.04$, $MAPE = 2.04\% \pm 0.02\%$. All of the scores, except the score for the total torque, are above the set expected threshold of $R^2 \geq 0.9$, $MAPE \leq 2.0\%$, indicating that they are high-quality models. The total torque achieves somewhat poorer results, but could still be usable in practice. This means that the goal of developing an ML system for predicting the torque values of a robotic manipulator was successful. Additionally, the fact that the models were possible to regress with a low standard error across folds, and that the generated dataset outputs have smooth distributions, indicates that a synthetic dataset can be used to regress this kind of problem.

The advantages of the used approach for modeling the torque are that the modeling process is less error-prone and user time-intensive in comparison with the classical methods. Still, it is not as precise as deterministically determining the torque model and requires a relatively powerful machine to be developed as the used neural networks are relatively large. Of course, it has to be noted again that, in a realistic application, data used would not be fully synthetic, but consist of either a mix of collected and synthetic data or only collected data. Limitations of the approach are clear, as the models developed are only valid for the used industrial manipulator and the modeling process would have to be repeated for different robots. Still, the approach could be implemented in cases of geometrically complex manipulators, especially ones with a higher number of degrees of freedom, in such applications where a precise torque value is not necessary.

As for the synthetic dataset generation, a number of applications are possible, which can be seen from the current research. It has to be noted that such data could have differences compared with real data, either due to modeling errors or outside influences. Still, if the process is verified, synthetic data generation can be used to generate new or additional data points and expand the collected datasets, especially in cases where the data collection is expensive or extremely time-consuming.

Future work in the field of dynamics modeling can rely on the process of generalizing the models to multiple manipulators, especially similar ones, through the introduction of additional input variables that pertain to the models in question, such as the mass and geometry of the manipulator links. Additional network architectures, such as LSTM networks, should also be tested, as they may be capable of fitting the data provided better. In the case of synthetic dataset generation, future work relating to the dynamics data being generated could focus on stricter comparisons of the synthetic data to the collected data in order to determine the possible statistical differences between the generated sets.

**Author Contributions:** Conceptualization, S.B.Š. and N.A.; methodology, S.B.Š., N.A., M.Š. and H.M.; software, S.B.Š.; validation, N.A., M.Š. and H.M.; formal analysis, M.Š. and H.M.; investigation, S.B.Š. and N.A.; resources, S.B.Š.; data curation, N.A.; writing—original draft preparation, S.B.Š. and N.A.; writing—review and editing, M.Š. and H.M.; visualization, S.B.Š.; supervision, M.Š. and H.M.; project administration, M.Š. and H.M.; funding acquisition, N.A., M.Š. and H.M. All authors have read and agreed to the published version of the manuscript.

**Funding:** This research received no external funding.

**Institutional Review Board Statement:** Not applicable.

**Informed Consent Statement:** Not applicable.

**Data Availability Statement:** The equations obtained from the described procedure, as well as the generated dataset, may be obtained through contact with the first author.

**Acknowledgments:** This research has been (partly) supported by the CEEPUS network CIII-HR-0108, European Regional Development Fund under the grant KK.01.1.1.01.0009 (DATACROSS), project CEKOM under the grant KK.01.2.2.03.0004, Erasmus+ project WICT under the grant 2021-1-HR01-KA220-HED-000031177, University of Rijeka scientific grant uniri-tehnic-18-275-1447, and project Metalska jezgra Čakovec (KK.01.1.1.02.0023).

**Conflicts of Interest:** The authors declare no conflict of interest.

**Abbreviations**

The following abbreviations are used in this manuscript:

| | |
|---|---|
| AI | Artificial Intelligence |
| ANN | Artificial Neural Network |
| DH | Dennavit–Hartenberg |
| LBFGS | Limited-Memory Broyden–Fletcher–Goldfarb–Shanno |
| LE | Lagrange–Euler |
| LSTM | Long short-term memory |
| MAPE | Mean Absolute Percentage Error |
| ML | Machine Learning |
| LSTM | Long short-term memory |
| MAPE | Mean Absolute Percentage Error |
| ML | Machine Learning |
| MLP | Multilayer Perceptron |
| NE | Newton–Euler |
| RS | Random Search |
| $R^2$ | Coefficient of determination |
| SGD | Stochastic Gradient Descent |

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
