# Peer review of "Dynamics Modeling of Industrial Robotic Manipulators: A Machine Learning Approach Based on Synthetic Data"

_mathematics, doi:10.3390/math10071174_

Round 1
Reviewer 1 Report
Dear authors,
you have presented an interesting topic and the quality of the paper is on a high level. It is also nice to see a paper with a high level of English language. I can suggest the paper for publication but prior to that some minor inadequate described parts need to be improved:
- at the end of the introduction emphasize more what of your work is innovative and express an added value for the scientific comunity. Don't be afraid to emphahsise where you are unique and well positioned with your work.
- by describing the selection of 20.000 synthetic dataset you surely need to consider the boundary conditions of your robotic manipulator. Present them clearly in the paper.
- When readers need to overview the paper it is necessary to get the condensed info abut the research in the conclusions. Therefore, add there the crucial data obtained during the performed research to support with them the presented statements in the conclusions at a glance.
Other minor comments are in the attached file.

Author Response
We would like to thank the first reviewer for providing us with the comments regarding our manuscript. We have resolved the provided comments, and we hope that our manuscript has been improved to a satisfactory degree. The answers to reviewer comments follow below.
- At the end of the introduction emphasize more what of your work is innovative and express an added value for the scientific comunity. Don't be afraid to emphahsise where you are unique and well positioned with your work.
The following text was added to the end of the introduction:
“The novelty presented by this paper is also two-fold - as there is a lack of similar research in both the modeling of dynamic models using regressive ML methods, and the creation and application of syntetic datasets for the given purpose. These contributions may allow researchers to simplify the process of dynamic modelling, or modelling in general - provided they have means to collect or synthetize the data.”
- By describing the selection of 20.000 synthetic dataset you surely need to consider the boundary conditions of your robotic manipulator. Present them clearly in the paper.
The following description was added.
“The ranges of the variables for random generation are set as given in Table 1. The values for the individual joints have been selected according to the ranges provided by the manufacturer [40]. Values for the minimal and maximal joint speeds and accelerations have been set uniformly for all joints, with the values selected as being realistic speeds and accelerations that could be encountered during the operation of the industrial robotic manipulator, to the ranges of [-1,1] rad/s and [-1,1] rad/s2”
- When readers need to overview the paper it is necessary to get the condensed info abut the research in the conclusions. Therefore, add there the crucial data obtained during the performed research to support with them the presented statements in the conclusions at a glance.
Due to the comments from both reviewers, the conclusion was expanded and rewritten to the following text:
“The paper presents the utilization of NE and LE algorithms for the model-ing of the industrial robot manipulator dynamics. The obtained models are then used to generate a synthetic dataset used for the training of ML-based models using the MLP algorithm. The achieved results are promising and point towards two possibilities. The first is the use of ML algorithms, namely ANNs, for the dynamic modeling of industrial robotic manipulators. It should be noted that in a realistic application scenario the data used would be collected from sensors. This leads to the second possibility investigated in the paper, which is the use of the synthetically designed dataset in the area of robotics modeling, which can assist in saving time and funds during research operations. Future work in the field could include the application of different ML al gorithms with the goal of model quality improvement, and further testing on synthetic datasets in robotics - such as investigating whether an improvement can be seen when real-world data is mixed with synthetically generated data. The paper presents the utilization of NE and LE algorithms for the dynamics modeling process of an industrial robotic manipulator. The paper also showcases the use of the generated models in the creation of a synthetic dataset, which is used to train an ML-based MLP algorithm. The torsion values have been regressed for each of the six joints, as well as the total torque. For the first joint the MLP managed to achieve a model with the scores of R2 = 0.96±0.01, M AP E = 1.18% ± 0.04%. Scores for the second joint were R2 = 0.09 ± 0.04, M AP E = 1.15% ± 0.03%, and for the third R2 = 0.95 ± 0.04, M AP E = 1.59% ± 0.03%. The scores for the fourth and fifth joint were R2 = 0.96 ± 0.03, M AP E = 1.81% ± 0.08% and R2 = 0.92 ± 0.05, M AP E = 1.91% ± 0.02%, respectively. The best-achieved scores for the sixth joint were R2 = 0.92 ± 0.03, M AP E = 1.93% ± 0.03%. Finally, the scores for the total torque of the industrial robotic manipulator were R2 = 0.89 ± 0.04, M AP E = 2.04% ± 0.02%. All of the scores, except the score for the total torque, are above the set expected threshold of R2 ≥ 0.9, M AP E ≤ 2.0%, indicating that they are high-quality models. The total torque achieves somewhat poorer results, but could still be usable in practice. This means that the goal of developing an ML system for predicting the torque values of a robotic manipulator was successful. Additionally, the fact that the models were possible to regress with a low standard error across folds, and that the generated dataset outputs have smooth distributions, indicates that a synthetic dataset can be used to regress this kind of problem.
The advantages of the used approach for modeling the torque are that the modeling process is less error-prone and user time-intensive in comparison to the classical methods. Still, it is not as precise as a deterministically determining the torque model and does require a relatively powerful machine to be developed - as the used neural networks are relatively large. Of course, it has to be noted again that, in a realistic application data used would not be fully synthetical - but consist of either a mix of collected and synthetic data or only the collected data. Limitations of the approach are clear, as the models developed are only valid for the used industrial manipulator, and the modelling process would have to be repeated for different robots. Still, the approach could be implemented in cases of geometrically complex manipulators, especially ones with a higher number of degrees of freedom, in such applications where precise torque value is not necessary. As for the synthetic dataset generation, a number of applications are possible, which can be seen from the current research. It has to be noted that such data could have differences in comparison to real data, either due to modelling errors or outside influences. Still, if the process is verified, the synthetic data generation can be used to generate new or additional data points, increasing the collected datasets - especially in cases where the data collection is expensive or extremely time-consuming.
Future work in the field of dynamics modeling can rely on the process of generalizing the models to multiple manipulators, especially similar ones, through the introduction of additional input variables that pertain to the models in question, such as the mass and geometry of the manipulator links. Additional network architectures, such as LSTM networks should also be tested, as they may be capable of fitting the data provided better. In the case of synthetic dataset generation, future work relating to the dynamics data being generated could focus on stricter comparisons of the synthetic data to the collected data, in order to determine the possible statistical differences between the generated sets.”
- Other minor comments are in the attached file.
All the minor comments, including unexplained abbreviations, manufacturer details, and others have been resolved in the reviewed manuscript.
Kindest regards,
Authors

Reviewer 2 Report
This work presents the modeling of the industrial robot manipulator dynamics, then the mathematical models are used to generate a synthetic dataset used for the training of Machine Learning-based models. The achieved results are used for the dynamic modeling of industrial robotic manipulators.
The provided information is relevant for the knowledge field. Nevertheless, some issues should be addressed before this manuscript could be considered for publication.
1) The Conclusion section is superficial, should include quantitative results, advantages and disadvantages, limitation and recommendation for real implementations, and future work should be extended.
2) For dynamical systems, recurrent neural networks are better suited than multilayer perceptron , recurrent neural network could be a good choice for future work.
Author Response
We would also like to thank the second reviewer for their review of our manuscript and the suggestions for future work. We have corrected the manuscript according to the provided comments and we hope that the manuscript is now at a satisfactory level. Please find the answers to the posed comments below.
- The Conclusion section is superficial, should include quantitative results, advantages and disadvantages, limitation and recommendation for real implementations, and future work should be extended.
Due to the comments from both reviewers, the conclusion was expanded and rewritten to the following text:
“The paper presents the utilization of NE and LE algorithms for the model-ing of the industrial robot manipulator dynamics. The obtained models are then used to generate a synthetic dataset used for the training of ML-based models using the MLP algorithm. The achieved results are promising and point towards two possibilities. The first is the use of ML algorithms, namely ANNs, for the dynamic modeling of industrial robotic manipulators. It should be noted that in a realistic application scenario the data used would be collected from sensors. This leads to the second possibility investigated in the paper, which is the use of the synthetically designed dataset in the area of robotics modeling, which can assist in saving time and funds during research operations. Future work in the field could include the application of different ML al gorithms with the goal of model quality improvement, and further testing on synthetic datasets in robotics - such as investigating whether an improvement can be seen when real-world data is mixed with synthetically generated data. The paper presents the utilization of NE and LE algorithms for the dynamics modeling process of an industrial robotic manipulator. The paper also showcases the use of the generated models in the creation of a synthetic dataset, which is used to train an ML-based MLP algorithm. The torsion values have been regressed for each of the six joints, as well as the total torque. For the first joint the MLP managed to achieve a model with the scores of R2 = 0.96±0.01, M AP E = 1.18% ± 0.04%. Scores for the second joint were R2 = 0.09 ± 0.04, M AP E = 1.15% ± 0.03%, and for the third R2 = 0.95 ± 0.04, M AP E = 1.59% ± 0.03%. The scores for the fourth and fifth joint were R2 = 0.96 ± 0.03, M AP E = 1.81% ± 0.08% and R2 = 0.92 ± 0.05, M AP E = 1.91% ± 0.02%, respectively. The best-achieved scores for the sixth joint were R2 = 0.92 ± 0.03, M AP E = 1.93% ± 0.03%. Finally, the scores for the total torque of the industrial robotic manipulator were R2 = 0.89 ± 0.04, M AP E = 2.04% ± 0.02%. All of the scores, except the score for the total torque, are above the set expected threshold of R2 ≥ 0.9, M AP E ≤ 2.0%, indicating that they are high-quality models. The total torque achieves somewhat poorer results, but could still be usable in practice. This means that the goal of developing an ML system for predicting the torque values of a robotic manipulator was successful. Additionally, the fact that the models were possible to regress with a low standard error across folds, and that the generated dataset outputs have smooth distributions, indicates that a synthetic dataset can be used to regress this kind of problem.
The advantages of the used approach for modeling the torque are that the modeling process is less error-prone and user time-intensive in comparison to the classical methods. Still, it is not as precise as a deterministically determining the torque model and does require a relatively powerful machine to be developed - as the used neural networks are relatively large. Of course, it has to be noted again that, in a realistic application data used would not be fully synthetical - but consist of either a mix of collected and synthetic data or only the collected data. Limitations of the approach are clear, as the models developed are only valid for the used industrial manipulator, and the modelling process would have to be repeated for different robots. Still, the approach could be implemented in cases of geometrically complex manipulators, especially ones with a higher number of degrees of freedom, in such applications where precise torque value is not necessary. As for the synthetic dataset generation, a number of applications are possible, which can be seen from the current research. It has to be noted that such data could have differences in comparison to real data, either due to modelling errors or outside influences. Still, if the process is verified, the synthetic data generation can be used to generate new or additional data points, increasing the collected datasets - especially in cases where the data collection is expensive or extremely time-consuming.
Future work in the field of dynamics modeling can rely on the process of generalizing the models to multiple manipulators, especially similar ones, through the introduction of additional input variables that pertain to the models in question, such as the mass and geometry of the manipulator links. Additional network architectures, such as LSTM networks should also be tested, as they may be capable of fitting the data provided better. In the case of synthetic dataset generation, future work relating to the dynamics data being generated could focus on stricter comparisons of the synthetic data to the collected data, in order to determine the possible statistical differences between the generated sets.”
- For dynamical systems, recurrent neural networks are better suited than multilayer perceptron, recurrent neural network could be a good choice for future work.
We would like to thank the reviewer for this suggestion. This was noted in future work part of the conclusion, and will be considered by authors going forward.
Kindest regards,
Authors
